# MolGround: A Benchmark for Molecular Grounding

## Abstract

Current molecular understanding approaches predominantly focus on the descriptive aspect of human perception, providing broad, topic-level insights. However, the referential aspect, i.e., linking molecular concepts to specific structural components, remains largely unexplored. To address this gap, we propose a molecular grounding benchmark designed to evaluate a model's referential abilities. This benchmark emphasizes fine-grained understanding and interpretability, challenging models to answer queries such as "What?", "Where?", and "Which ones?" across various cognitive levels. We align molecular grounding with established conventions in NLP, cheminformatics, and molecular science, showcasing the potential of recent NLP techniques to advance molecular understanding within the AI for Science movement. Specifically, we introduce the largest molecular grounding benchmark to date, consisting of 187k QA pairs across five tasks, each targeting a distinct cognitive level. Extensive evaluations of both general-purpose and domain-specific (M)LLMs highlight the challenges posed by this benchmark. While existing techniques, such as in-context learning, fine-tuning, and multi-agent strategies, can improve performance, significant progress is still needed to enhance referential capabilities. Furthermore, we demonstrate that molecular grounding can also benefit traditional tasks such as molecular captioning and Anatomical, Therapeutic, Chemical (ATC) classification. The source code and data are available at https://anonymous.4open.science/r/MolGround-2025/.

## 1 Introduction

Deep learning models have transformed traditional molecular understanding tasks, including property prediction Wu et al. (2017); Zaidi et al. (2023); Zhang et al. (2024b), molecular generation Xu et al. (2019); Fang et al. (2024b); Song et al. (2024), and reaction prediction Fooshee et al. (2018); Chen et al. (2025); Tavakoli et al. (2024). Recently, tasks like molecular captioning Edwards et al. (2021); Zhang et al. (2025) and molecule-language translation Edwards et al. (2024) have gained significant attention due to advancements in large language models Li et al. (2024); Pei et al. (2024). These models represent molecular structures as sequences of tokens, enabling the generation of natural language descriptions by leveraging sophisticated sequence-to-sequence learning techniques.

While having yielded promising results, these approaches primarily mimic the **descriptive** aspect of human perception Cocchiarella (1974); Geach (1950); Kamp & Reyle (1993), focusing on broad, topic-level understanding. The **referential** aspect of perception, which associates concepts with specific molecular components (e.g., atoms, functional groups, rings), has been overlooked. For example, consider the SMILES *CC(=O)O (acetic acid)*. In molecular captioning, a typical output might be: "This is acetic acid, commonly known as the main component of vinegar. It is used industrially in production and exhibits toxic effects at high concentrations." While this description is highly informative, it is descriptive in nature. From a referential perspective, it is more critical to identify which specific part of the molecule contributes to its toxicity. In this case, the *carbonyl group (C=O)* is responsible for the molecule's corrosive effects, as it facilitates the release of protons *(H+)*, which can damage biological tissues. This referential understanding not only enhances interpretability but also generalizes to other similar compounds, such as *formic acid C(=O)O*, *oxalic acid C(=O)(O)C(=O)O*, and *trichloroacetic acid C(Cl)(Cl)(Cl)(=O)O*.

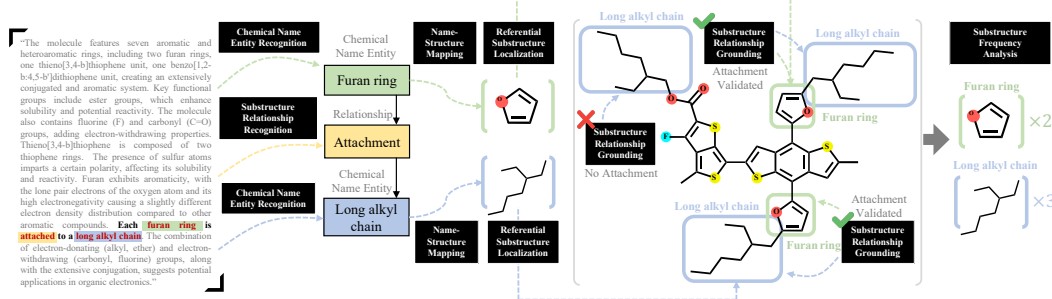

Figure 1: A multimodal referential framework for fine-grained molecular grounding, comprising five tasks: Chemical Name Entity Recognition, Name-Structure Mapping, Referential Substructure Localization, Substructure Relationship Grounding, and Substructure Frequency Analysis, demonstrated through a running example.

While the complementary nature of descriptive and referential perceptions has long been modeled in cognitive science, such as in Fregean Semantics Cocchiarella (1974), Russell's Theory of Descriptions Geach (1950), and Discourse Representation Theory (DRT) by Hans Kamp and Uwe Reyle Kamp & Reyle (1993), it has also been successfully implemented recently in vision-language research Deitke et al. (2025); Arai & Tsugawa (2024); Liu et al. (2023). The integration of visual grounding Xiao et al. (2024); Deng et al. (2021), which mimics referential perception by linking textual concepts to specific image regions, has significantly advanced the performance, interpretability, and generalization of vision-language models. These models, which traditionally relied solely on image-caption pairs for training, have greatly benefited from this approach.

Believing that molecular understanding research is at a similar turning point, we propose a grounding benchmark to assess a model's ability to explicitly associate molecular concepts with specific structural components. This benchmark emphasizes fine-grained understanding and interpretability, enabling models to identify, explain, and reason about the roles of particular molecular features. Unlike visual grounding, where a model is primarily tasked with identifying the locations of concepts, molecular grounding requires the identification of specified components at multiple cognitive levels, including concept instances, structural locations, and compositional facts. From a semantic interpretation perspective, molecular grounding differs from existing molecular understanding tasks that focus on general and broad content interpretation. Instead, it emphasizes providing answers to fine-grained queries such as "What are the contextual entities?", "Where are they?", and "Which ones?". Figure 1 illustrates the proposed molecular grounding tasks including Chemical Name Entity Recognition (CNER), Name-Structure Mapping (NSM), Referential Substructure Localization (RSL), Substructure Relationship Grounding (SRG), and Substructure Frequency Analysis (SFA). This paper serves as a pilot study aimed at formulating molecular grounding by aligning it with established conventions in NLP, cheminformatics, and molecular science. Our findings demonstrate that NLP techniques can play a critical role in advancing molecular understanding within the broader AI for Science movement. In addition to creating the largest molecular understanding benchmark to date and conducting extensive evaluations on existing (M)LLMs and NLP techniques, we also demonstrate that the grounding molecular can be successfully integrated to enhance conventional tasks like molecular captioning and ATC (anatomical, therapeutic, chemical) classification. Our findings underscore the importance of referential understanding in improving interpretability, generalization, and practical applications in molecular science.

## 2 RELATED WORK

Molecular understanding has been a long-standing field of research, predating the recent surge of interest in AI for Science. The tasks in this field can be broadly grouped into three categories based on their popularity: 1) Property prediction Wu et al. (2017); Walters & Barzilay (2021); Zhang et al. (2024a) and representation learning Xiaomin Fang & Wang (2022); Zhang et al. (2024b), which are the most extensively studied and widely popular. 2) Structure prediction Jumper et al. (2021); Song et al. (2024), captioning Li et al. (2024); Edwards et al. (2021), and generation Xu et al. (2019);

Hua et al. (2024), which have recently gained significant attention. 3) Emerging studies on tasks such as reaction prediction and optimization Fooshee et al. (2018), interaction prediction Tavakoli et al. (2024), simulations and dynamics Vander Meersche et al. (2024), toxicity and safety assessment Sahu & Poler (2024), and visualization and explainability Janissen et al. (2024). The popularity of the first two groups largely stems from the ease of directly applying sophisticated machine learning models to these tasks. Early approaches relied on methods like Bayesian classifiers Langley et al. (1992), logistic regression Hosmer Jr et al. (2013), and SVMs Hearst et al. (1998), while more recent efforts have widely adopted CNNs O'Shea (2015), GNNs Wu et al. (2020), and Transformer-based models Vaswani (2017). Most of these models follow a pipeline of encoding molecules into embeddings and predicting outputs such as labels or textual descriptions, reflecting the way these models were initially designed. However, these implementations tend to be **descriptive**, as they focus on high-level concepts by treating a molecule as a whole, rather than addressing its internal components. The third group, on the other hand, signals a shift toward more fine-grained modeling and improved interpretability. This shift is driven by two factors: 1) The needs of identifying subcomponents in molecular science, such as reaction tracing Smith & March (2007); Lei Fang & Lou (2022); Umit V. Ucak (2021) and understanding molecule-target interactions Lipinski et al. (1997); Segler et al. (2018). 2) Advancements in machine learning for interpretability and generalization Gao & Guan (2023); Zhang et al. (2025). Ultimately, these developments highlight the growing demand for models with **referential** perception, enabling them to go beyond high-level descriptions and address specific components within a molecule.

Table 1: QA pair distribution comparison across descriptive (Des.) and referential (Ref.) perceptions with existing molecular understanding benchmarks (ChemBench4K(CB4) Zhang et al. (2024a) and MoleculeQA (MQA) Lu et al. (2024)).

| Bench. | Tasks | #QA | Des.% | Ref.% | Bench. | Tasks | #QA | Des.% | Ref.% |
|--------|-------|-----|-------|-------|--------|-------|-----|-------|-------|
| CB4 | Caption2mol | 800 | 97.75 | 2.25 | MQA | Property | 6,267 | 100.0 | 0.00 |
| | Mol2Caption | 299 | 100.0 | 0.00 | | Usage | 3,074 | 100.0 | 0.00 |
| | Name Conv. | 799 | 99.87 | 0.13 | | Source | 13,630 | 100.0 | 0.00 |
| | Prod. Pred. | 300 | 96.99 | 3.01 | | Structure | 38,603 | 83.62 | 16.38 |
| | Yield Pred. | 300 | 100.0 | 0.00 | | Total | 61,574 | 91.19 | 8.81 |
| | Temp. Pred. | 202 | 98.98 | 1.02 | **MolGround** (ours) | CNER | 11,185 | 0.00 | 100.0 |
| | Solv. Pred. | 300 | 87.66 | 12.34 | | BNSM | 4,431 | 100.0 | 0.00 |
| | Retrosynth. | 300 | 96.00 | 4.00 | | RSL | 53,827 | 0.00 | 100.0 |
| | Prop. Pred. | 709 | 59.23 | 40.77 | | SRG | 82,132 | 0.00 | 100.0 |
| | Total | 4,009 | 90.88 | 9.12 | | SFA | 35,992 | 0.00 | 100.0 |
| | | | | | | Total | **187,567** | 0.02 | **99.98** |

The complementary relationship between descriptive and referential perceptions has been extensively explored in cognitive science, as seen in Fregean Semantics (senses and references) Cocchiarella (1974), Russell's Theory of Descriptions (definite descriptions and proper names) Geach (1950), and Discourse Representation Theory (descriptions and referents) Kamp & Reyle (1993). However, referential perception has not been explicitly modeled or systematically evaluated in molecular understanding. From this perspective, Table 1 summarizes commonly adopted benchmarks like ChemBench4K Zhang et al. (2024a), and MoleculeQA Lu et al. (2024). It is clear that this is an area requiring more focused and explicit attention. While recent advancements in models have made strides toward incorporating referential perception, with promising results observed in integrating referential perception-oriented visual grounding (e.g., RefFormer Wang et al. (2024), ClawMachine Ma et al. (2024), DOrA Wu et al. (2024)), significant challenges remain. This results from the heavy reliance on costly human expertise for benchmark construction and the lack of a systematic formulation of the problem. As highlighted in Table 1 , our proposed MolGround represents an initial effort to address these challenges. It scales up to 3 times larger than existing benchmarks and introduces fine-grained definitions to better align with the requirements of referential perception.

## 3 MOLECULAR GROUNDING TASKS

We define five groups of grounding tasks by aligning to the common conventions in NLP, cheminformatics, and molecular science. The alignment and challenges of each task are summarized in Table 2.

Table 2: Grounding tasks with Bloom's Cognitive Levels, corresponding challenges, and required abilities.

| Cognitive levels | Tasks | Challenges | Required Abilities |
|---|---|---|---|
| Remember | CNER | Diverse Entity Forms | Chemical Knowledge Recall; Syntax Understanding |
| Understand | BNSM | Multimodal Transformation | Semantic Understanding; Syntax Understanding |
| Apply | SFA-Atom | Multi-instances; Coreference Resolution | Substructure Matching; Pattern Recognition |
| Analyze | SFA-Heteroatoms Type | Similarity Differentiation | Categorization |
| | SFA-Monocyclic Ring Type | Structural Similarity | Pattern Recognition; Categorization |
| | SFA-Non-exist Ring | Absence Detection | Negative Pattern Recognition |
| | SRG | Multidimensional Relations | Structural Understanding; Relationship Inference |
| | Singular RSL | Multi-instances | Spatial Reasoning; Pattern Recognition |
| Evaluate | SFA-Ring | Multiple Representation forms | Structural Comparison; Quantitative Analysis |
| | SFA-Substructure | Structural Variability | Pattern Recognition; Logical Deduction |
| | Multiple RSL | Multiple Structures; Diverse Relation Types | Contextual Reasoning |

**Chemical Named Entity Recognition (CNER)**: *Recognize and extract chemical entity names (e.g., molecule names, substructure names, or functional groups) as a set $\mathcal{N}$ from a given caption $\mathcal{X}$ as*

$$f_N : \mathcal{X} \mapsto \mathcal{N}$$
$$\mathcal{X} = \{x_i\}, \mathcal{N} = \{n_j\} \tag{1}$$

where $x_i$ is the $i^{th}$ token in the sequence $\mathcal{X}$, and $n_j$ is a quadruple $(c_j, b_j, l_j, r_j)$ consisting of the $j^{th}$ extracted name entity $c_j$ and its role $r_j \in \mathcal{R}$, beginning position $0 \leq b_j \leq \|X\|$, and length $l_j$. Note $\mathcal{R}$ is a set of predefined roles (e.g., donor, acceptor) which are contextual and application-specific.

This task reflects referential perception by linking textual mentions of chemical entities to their semantic roles and serves as a foundation for molecular grounding by identifying key entities for downstream tasks. While similar to Named Entity Recognition (NER) in NLP, CNER extends the task by additionally identifying the roles of extracted entities. Furthermore, unlike NER, where entities are typically proper nouns or noun phrases, chemical entities are significantly more diverse and technically complex. For instance, the *drug acetaminophen* exemplifies this complexity: it has multiple IUPAC names, such as *N-(4-hydroxyphenyl)acetamide*, *4'-hydroxyacetanilide*, and *p-hydroxyacetanilide*; a molecular formula, *C8H9NO2*; an InChI representation, *InChI=1S/C8H9NO2/c1-6(10)9-7-2-4-8(11)5-3-7/h2-5,11H,1H3,(H,9,10)*; as well as various SMILES representations and trade names like *Tylenol*, *Panadol*, and *Calpol*. As illustrated in Figure 2a, a CNER model must accommodate these diverse forms, demanding the ability to recall chemical domain knowledge and a deep understanding of chemical syntax and representation conventions.

**Bidirectional Name-Structure Mapping (BNSM)**: *Translate chemical names $\mathcal{N}$ into corresponding structural representations (e.g., SMILES, InChI, molecular graphs) $\mathcal{S}$ or convert given structural representations back into their corresponding names as*

$$f_{n2s} : \mathcal{N} \mapsto \mathcal{S}$$
$$f_{s2n} : \mathcal{S} \mapsto \mathcal{N}$$
$$\mathcal{N} = \{n_i\}, \mathcal{S} = \{s_j\} \tag{2}$$

where structural representation $\mathcal{S}$ is sequences of textual codes in SMILES, InChI, or molecular graphs wrapping atoms (nodes) and bonds (edges).

This task bridges textual and structural representations, embodying referential perception by grounding a molecule's name to its physical structure and vice versa. It aligns with semantic-vision aliment tasks in multimedia and structure-based prediction tasks in cheminformatics. As illustrated in Figure 2a, unlike the sequence-to-sequence framework used in semantic-vision aliment, this task introduces an additional multimodal challenge. This complexity arises from the hierarchical and graph-based nature of molecular structures, which are governed by spatial and chemical constraints. Furthermore, this task exhibits extremely low error tolerance, as even a minor mistake in structural representation can result in a fundamentally different molecule. For instance, the molecules represented by *C1=CC=CC=C1* and *C1=NC=CC=C1* differ by only a single atom, yet correspond to entirely distinct chemical entities.

**Referential Substructure Localization (RSL)**: *Identify the specific occurrences of substructures (e.g., functional groups, rings, or chains) within a molecule's structural representation $\mathcal{G}$, based on*

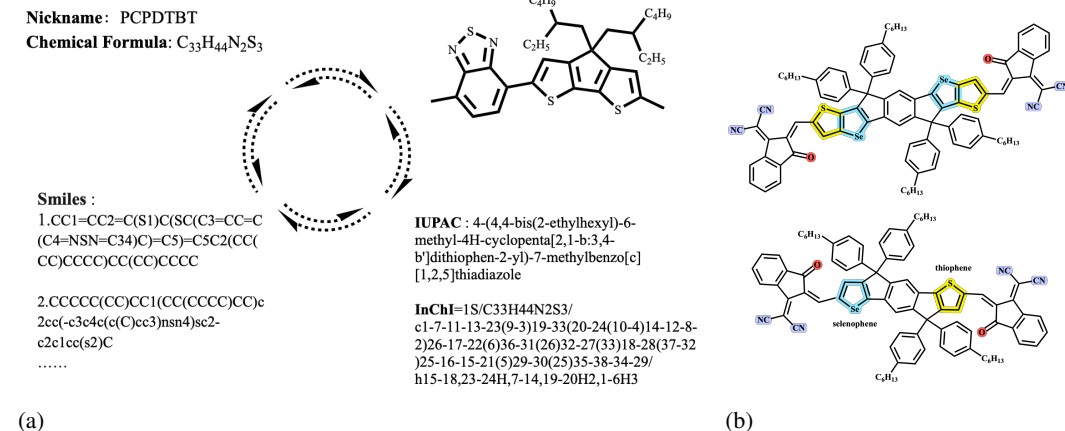

**Nickname**: PCPDTBT
**Chemical Formula**: $C_{33}H_{44}N_2S_3$

**Smiles**:
1. CC1=CC2=C(S1)C(SC(C3=CC=C(C4=NSN=C34)C)=C5)=C5C2(CC(CC)CCCC)CC(CC)CCCC

2. CCCCC(CC)CC1(CC(CCCC)CC)c2cc(-c3c4c(c(C)cc3)nsn4)sc2-c2c1cc(s2)C
......

**IUPAC**: 4-(4,4-bis(2-ethylhexyl)-6-methyl-4H-cyclopenta[2,1-b:3,4-b']dithiophen-2-yl)-7-methylbenzo[c][1,2,5]thiadiazole

**InChI**=1S/C33H44N2S3/c1-7-11-13-23(9-3)19-33(20-24(10-4)14-12-8-2)26-17-22(6)36-31(26)32-27(33)18-28(37-32)25-16-15-21(5)29-30(25)35-38-34-29/h15-18,23-24H,7-14,19-20H2,1-6H3

(a)            (b)

Figure 2: Challenges of the proposed grounding tasks. (a) Diversity in naming conventions and multimodal gaps between textual and structural representations. (b) Two rings, e.g., the thiophene ring (yellow) and the selenophene ring (blue) which share a "functional integration" relationship, are not necessarily physically adjacent.

*their names or descriptions $\mathcal{N}$ as*

$$f_L : (\mathcal{N}, \mathcal{G}) \mapsto \mathcal{L}$$
$$\mathcal{N} = \{n_i\}, \; \mathcal{G} = (\mathcal{V}, \mathcal{E}), \; \mathcal{G}_i \subseteq \mathcal{G},$$
$$\mathcal{L} = \{\mathcal{L}_i\} \in \{\mathcal{G}_i\} \times \mathcal{G} \quad\quad (3)$$

where $\mathcal{V}$ is the set of atoms (nodes) and $\mathcal{E}$ is the set of bonds (edges), $\mathcal{G}_i = (\mathcal{V}_i, \mathcal{E}_i)$ is the substructure graph for $n_i$, and $\mathcal{L}_i$ is the location indicator for $\mathcal{G}_i$ within the molecular graph $\mathcal{G}$. $\mathcal{L}_i = (\mathcal{L}_i^{atom}, \mathcal{L}_i^{bond})$ consists of indices of $\mathcal{G}_i$'s atoms and bonds within the molecular graph $\mathcal{G}$, where $\mathcal{L}_i^{atom} = \{m | v_m \in \mathcal{V}_i\}$ and $\mathcal{L}_i^{bond} = \{(m, n) | (v_m, v_n) \in \mathcal{E}_i\}$.

This task emphasizes referential perception by mapping textual or conceptual references to their precise structural counterparts. It is analogous to object detection in vision and token-level alignment in NLP. Building upon CNER and BNSM, the new challenge imposed in RSL is the existence of multiple instances of the target and possible distractors. Those distractors are often with similar structures as the target, further challenging the low tolerance at fine grained level. Examples can be found in Figure 2b. *Selenophene* rings, differing by only one atom from *thiophene*, may further complicate localization.

**Substructure Relationship Grounding (SRG)**: *Identify the relationships (e.g., composition, directed attachment or functional integration) between substructures within a molecule, as represented by a caption $\mathcal{X}$ and the corresponding molecular graph $\mathcal{G}$ as*

$$f_K : (\mathcal{X}, \mathcal{G}) \mapsto \mathcal{K}$$
$$\mathcal{X} = \{x_i\}, \; \mathcal{G} = (\mathcal{V}, \mathcal{E}), \; \mathcal{G}_i, \mathcal{G}_j \subseteq \mathcal{G},$$
$$\mathcal{K} = \{k_{ij}\} \in \{\mathcal{G}_i\} \times \{\mathcal{G}_j\} \quad\quad (4)$$

where $\mathcal{G}_i$ and $\mathcal{G}_j$ are the $i^{th}$ and $j^{th}$ substructure graphs, and $k_{ij}$ is their identified relationship.

This task builds on referential perception by modeling the interactions and dependencies between molecular substructures, providing insights into their functional roles. It draws parallels to object relation analysis in image and video scene understanding, and interaction modeling in molecular sciences. The key challenge in this task lies in the multidimensional nature of the relationships. Unlike conventional visual question answering or image-text alignment—where relationships are often defined through semantic or co-occurrence-based correlations—SRG relationships are multidimensional, incorporating chemical, spatial, physical, and hierarchical factors. More specifically, this complexity means that chemical relationships, such as composition, directed attachment, or functional integration, are intricately intertwined with their associated physical factors. This contrasts sharply with common semantic relationships in NLP or vision-language tasks (e.g., is-a, is-part-of), which are often straightforwardly defined. Figure 2b illustrates this challenge.

**Substructure Frequency Analysis (SFA)**: *Count the number of occurrences of a specified substructure (indicated by its name $n_i \in \mathcal{N}$) within the structural representation $\mathcal{G}$ of a given molecule as*

$$
\begin{aligned}
f_F : (\mathcal{N}, \mathcal{G}) &\mapsto \mathcal{F} \\
\mathcal{N} = \{N_i\}, \; \mathcal{G} &= (\mathcal{V}, \mathcal{E}), \; \mathcal{G}_i \subseteq \mathcal{G}, \\
\mathcal{F} &= \{f_i\} \in \mathbb{N}
\end{aligned}
\tag{5}
$$

where $f_i$ is the frequency of $n_i$ counted by the occurrences of its substructure graph $\mathcal{G}_i$ within $\mathcal{G}$.

This task extends referential perception by quantifying the presence of referenced substructures, supporting downstream molecular grounding tasks such as property prediction or functional analysis. It aligns with object counting in multimedia and motif detection in cheminformatics. However, this goes beyond a simple counting task. The complexity arising from multiple representation forms, hierarchical definitions, multidimensionality, and multiresolution makes the target of counting dynamic and context-dependent.

# 4 BENCHMARKING

Benchmarking in the chemical domain is expensive, largely due to its heavy reliance on human expertise. To build the largest molecular understanding benchmark to date, we adopt an interactive approach based on the Spiral Model Boehm (1986). Specifically, we develop a prototype of a grounding agent to facilitate the process. The agent automates data collection, cleaning, and structuring, after which the data is validated, corrected, or filtered by human experts. Data entries rejected and refined by human experts are recorded by the agent and the agent updates the follow-up results for further review. This iterative interaction between humans and the agent continues until convergence is achieved. Throughout this process, the agent itself is iteratively improved as part of an exploration into effective grounding methodologies, while simultaneously enhancing both the scale and quality of the benchmark.

## 4.1 DATA COLLECTION AND PREPROCESSING

We collected molecules from existing molecular captioning datasets, such as ChEBI-20 Edwards et al. (2021) and LPM-24 Edwards et al. (2024). Additionally, we extended our collection with molecules published in chemical literature Nagasawa et al.. In total, this resulted in a dataset of 55,989 molecules. The collected molecules exhibit varying levels of structural complexity. Specifically, the number of atoms per molecule ranges from 1 to 574, with a median value of 33. The number of atom type is 70. The number of rings varies from 0 to 69, while the number of bonds spans from 0 to 642.

For molecules lacking captions, we utilized GPT-4o OpenAI & Josh Achiam (2024) to generate detailed captions. This was achieved by inputting the molecule's IUPAC name, SMILES representation, relevant literature, and molecular structure image into GPT-4o, along with prompt templates designed by chemical experts (details provided in Appendix E). The templates were tailored to generate fine-grained substructure-focused content, such as identifying the substructures within a molecule, describing how they are connected, and outlining their properties. Besides, we also include the conjugation system (such as $\pi - \pi$ bonds, $p - \pi$ atoms and aromatic systems) and stereochemistry (such as chiral centers) in the caption. As a result, we constructed a dataset of 55,989 molecule-caption pairs.

## 4.2 GROUNDING AGENT PROTOTYPING

We build a multi-agent system composed of an LLM-based text interpreter, a meta-retriever, a structure parser, and a structure memory bank to provide initial result and interact with human annotator for refinement. Specifically, the text interpreter leverages large language models (LLMs) to perform named entity recognition and relationship analysis based on the collected meta data. The meta-retriever is built using PubChem APIs Kim et al. (2024) and is responsible for collecting molecule and substructure entity information, e.g., names, properties, structures, and descriptions. The structure parser is developed using RDKit and handles substructure retrieval, comparison, and validation.

These three agents work collaboratively to perform structurization on each molecule and output initialized grounding results. More details of the prototype is included in Appendix B.

### 4.3 HUMAN ANNOTATION

The structuring and annotation process is performed iteratively, enabling the grounding agent to collaborate effectively with human chemical experts. Specifically, the structurization results are sequentially reviewed by seven human annotators for verification and refinement. If a result is rejected and refined by an annotator, the revised output is returned to the agent to generate updated results for the subsequent steps. This iterative process continues until all structurization outputs are reviewed and validated by the annotators. Six annotations worked on disjoint subsets, and all annotations were independently reviewed by two experts. The inconsistency rate between the initial annotations and expert verifications is 3.84%, and the final dataset is based on the consensus results agreed upon by both experts. Furthermore, the inter-expert agreement, measured using Cohen's kappa, is 0.985, indicating high consistency and annotation quality.

### 4.4 QA CONSTRUCTION AND BENCHMARK STATISTICS

After annotating each task, we use predefined templates (Appendix F) to convert the annotations into question–answer (QA) pairs. As a result, the MolGround benchmark contains 187,567 questions. Specifically, for each question, we provide both multi-choice and open-ended answer format. The number of questions for each task is shown in Table 1. Specifically, in order to evaluate the fine-grained referential ability, MolGround covers 307 fundamental substructures spanning a wide range of structural types (as shown in Figure 7 in Appendix), including atoms, rings (monocyclic, bicyclic, tricyclic), chains (straight, branched), functional groups, and composite units. Specifically, the fused unit collection consists of fused substructures composed of rings, chains, or functional groups that exhibit repeated patterns (e.g., terthiophene, which contains three thiophene rings connected via C–C bonds). Besides, we also include isomerism in the grounding tasks. We split the QA data into training, validation, and test sets using an 80%/10%/10% ratio for each task. There is no individual molecule appears in more than one split.

## 5 EXPERIMENTS

### 5.1 BASELINES

We employ five LLMs as baselines, including general-domain models like GPT-4o OpenAI & Josh Achiam (2024) and LLaMA 3.1 (8B and 70B) Grattafiori & Abhimanyu Dubey (2024), as well as models specifically tailored for molecular understanding, such as ChemLLM (7B) Zhang et al. (2024a) and Mol-Instructions Fang et al. (2024a). Furthermore, we investigate LLM learning techniques, including In-Context Learning (ICL), i.e., Few-shot learning, and Supervised Fine-tuning (SFT) using LoRA Hu et al. (2022). Given that molecular structures can be represented as graphs, we also incorporate Multi-modal LLMs (MLLMs) like GPT-4o Vision OpenAI & Josh Achiam (2024) and LLaVA-Next Liu et al. (2023) in our evaluations. The evaluation metrics are included in the Appendix G.

### 5.2 ZERO-SHOT PERFORMANCE OF LLMS AND MLLMS

Table 3a compares the performance of LLMs and MLLMs across five molecular grounding tasks. Specifically, we split RSL into singular- and multiple-substructure grounding tasks, where the former requires locating a single named entity mentioned in the caption, while the latter involves identifying all named entities. Overall, most tasks remain challenging for all baseline models, with accuracy generally below 0.5. RSL proves to be the most difficult, with all models achieving accuracies below 0.016. In contrast, SFA exhibits relatively better performance, with the highest F1-score (0.705) achieved by LLaMa3.1-70B. In addition, models perform well on counting non-existent rings but struggle with monocyclic ring identification due to subtle structural differences. In CNER, models excel at named entity recognition but fail to correctly assign roles to extracted chemical names. For BNSM, models perform well on SMILES-to-Formula translations but poorly

Table 3: Evaluation of the LLM and MLLMs on the proposed tasks. RSL is spited to singular substructure localization (S-RSL) and multiple substructures localization (M-RSL).

(a) Zero-shot performance comparison.

| Model | CNER | BNSM | SRG | SFA | S-RSL | M-RSL |
|---|---|---|---|---|---|---|
| **LLM Baselines** | | | | | | |
| GPT4o | 0.197 | **0.345** | **0.248** | 0.434 | 0.015 | 0.012 |
| LLaMa3.1-8B | 0.239 | 0.096 | 0.185 | 0.424 | 0.006 | 0.000 |
| LLaMa3.1-70B | **0.242** | 0.300 | 0.146 | **0.705** | 0.008 | 0.001 |
| chemllm-7B | 0.005 | 0.031 | 0.146 | 0.040 | 0.000 | 0.000 |
| Mol-Instructions | 0.130 | 0.021 | 0.012 | 0.075 | 0.000 | 0.000 |
| **MLLM Baselines** | | | | | | |
| GPT4o-Vision | 0.197 | **0.345** | 0.086 | 0.494 | **0.016** | **0.015** |
| LLava-Next | 0.016 | 0.146 | 0.067 | 0.398 | 0.010 | 0.001 |

(b) Performance with ICL, SFT, and Agents.

| Model | CNER | BNSM | SRG | SFA | S-RSL | M-RSL |
|---|---|---|---|---|---|---|
| **Baselines + ICL** | | | | | | |
| LLaMA3.1-8B | 0.022 | 0.276 | 0.309 | 0.552 | 0.003 | 0.120 |
| Mol-Instructions | 0.208 | 0.243 | 0.374 | 0.399 | 0.001 | 0.001 |
| **Baselines + SFT** | | | | | | |
| LLaMA3.1-8B | **0.938** | 0.231 | **0.998** | **0.964** | 0.275 | 0.315 |
| Mol-Instructions | 0.889 | 0.115 | 0.985 | 0.895 | 0.295 | 0.337 |
| **Grounding Agent** | | | | | | |
| GPT-4o | – | – | – | – | **0.630** | **0.541** |
| LLaMA3.1-8B | – | – | – | – | 0.334 | 0.426 |

on others (e.g., IUPAC-to-SMILES and nickname-to-InChI). Additionally, complex structure transformations are more challenging than simple mappings, often producing incorrect yet structurally similar names or structures. SRG results highlight difficulties in establishing substructure relationships, with models misinterpreting textual cues and neglecting structural connections. RSL shows particularly poor performance, with the best F1-score (0.016) achieved by GPT-4o-Vision. While models succeed in predicting the number of atom indices for a location, they fail to provide the correct indices. In addition, they struggle to locate ambiguous named entities (e.g., 'long alkyl chain') that adopt different structural forms in different molecules. MLLMs, which leverage structural images, outperform LLMs on most tasks, except SRG, where visual distractions hinder their ability to interpret relations between multiple instances.

## 5.3 EVALUATION OF ICL, SFT, AND MULTI-AGENT TECHNIQUES

Table 3b compares the effects of in-context learning (ICL) and supervised fine-tuning (SFT) on task performance. Overall, both ICL and SFT improve performance on several tasks, particularly those requiring chemical knowledge recall and pattern recognition (e.g., CNER and SFA). It is worth noting that SFT also brings significant improvement to the SRG when sufficient learning examples are provided. With task-specific training data, SFT yields higher performance gains than ICL on RSL. For instance, LLaMA3.1-8B's F1-score rises from 0.006 to 0.275 in singular RSL and from 0.001 to 0.315 in multiple RSL. However, these improvements are not consistent. Both ICL and SFT degrade performance on BNSM because structural and naming conversion cannot be easily transferred. For instance, for BNSM, likely because instance-level pairwise relationships depend not only on what the instances are, but also on where they are located. As illustrated in Figure 1, the relationship between the furan ring and the long alkyl chain is not always literal as described in the caption. It depends on the specific instance of the chain being referred to. For example, the long alkyl chain in the top-left corner is not connected to the furan ring.

We also evaluate the RSL performance of our grounding prototype for comparison. The grounding agents outperform baseline methods and ICL, SFT techniques in both singular and multiple RSL tasks. This advantage primarily comes from integrating a subgraph matching tool, which enables more precise location generation for queried chemical entities and consequently improves grounding accuracy. However, the grounding agents still show significant limitations. Figures 3 and 4 illustrate these issues. In Figure 3, the grounding result for thiophene is shown. While GPT-4o correctly identifies thiophene as a five-membered substructure, its predictions are scattered across multiple locations. The grounding agent achieves more accurate localization but fails to satisfy the constraint that thiophene is part of a thieno[3,2-b]thiophene moiety, as described in the caption. Figure 4 highlights a limitation of the subgraph retrieval technique applied to RSL. Here, the predicted locations of a chain are overlapped and scattered across different substructures, reflecting the model's difficulty in contextual disambiguation.

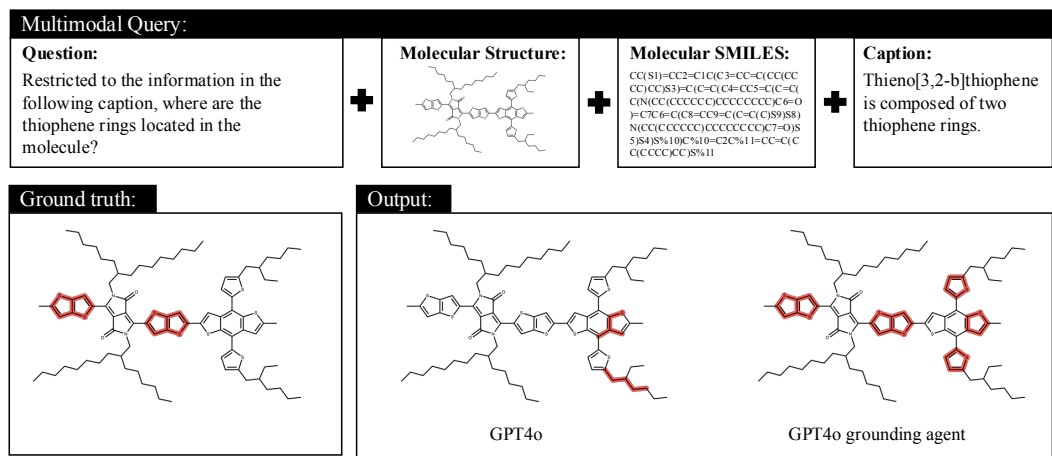

Figure 3: Comparison between the ground truth and grounding outputs by GPT-4o and the agent.

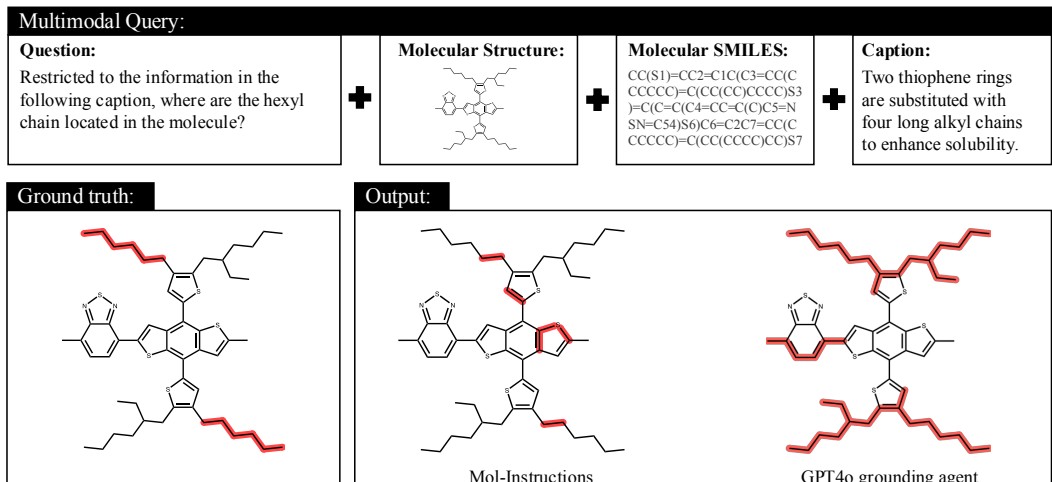

Figure 4: Irrelevant and wrong grounding results generated by Mol-Instructions and the agent.

## 5.4 CAN GROUND HELP DOWNSTREAM TASKS?

We conducted experiments to evaluate the impact of molecular grounding on molecular captioning and classification. For molecular captioning, we incorporate grounding results generated by the grounding agent (GPT4-based) as additional input. As shown in Table 4 (in Appendix H), this additional information improved performance across all BLEU metrics.

For molecular classification, we investigated the effectiveness of integrating molecular substructure information into ATC classification (Anatomical, Therapeutic, Chemical). We use ATC-CNN Cao et al. (2022) as the baseline and conduct experiments on ATC-SMILES Cao et al. (2022) dataset with the resulting substructures. As shown in Table 5 (in Appendix H), incorporating substructure information led to significant performance gains across almost all evaluation metrics, including aiming (+3.37%), coverage (+7.59%), accuracy (+4.26%), and absolute true (+1.38%), demonstrating that molecular grounding enhances drug classification.

## 6 CONCLUSION

This paper introduces a molecular grounding benchmark aimed at enhancing the referential aspect of molecular understanding. We present MolGround, a dataset comprising 187k QA pairs across five subtasks, which represents the largest fine-grained referential benchmark for molecular QA. Our evaluation shows that both general-domain and molecular large language models ((M)LLMs) struggle with these tasks, although supervised fine-tuning (SFT) and in-context learning (ICL) provide some improvements. While our multi-agent grounding prototype outperforms existing baselines, including SFT and ICL, these grounding tasks remain challenging and warrant further research effort.

## ETHICS STATEMENT

The primary contribution of this work is the introduction of a new benchmark dataset for molecular grounding, which aims to address the gap in referential aspects of molecular understanding. The data of our benchmark is derived from public domain chemical database, ensuring no privacy or licensing issues are involved. The research does not involve human subjects, and we foresee no direct potential for malicious use. Our goal is to provide a resource that upholds high standards of scientific excellence and promotes fair and transparent evaluation of model capabilities.

## REPRODUCIBILITY STATEMENT

The source code with the QA pair benchmark dataset, is made available at the following anonymized repository: https://anonymous.4open.science/r/MolGround-2025/. The appendix of the paper provides comprehensive details regarding the experimental setup. Furthermore, we provide detailed descriptions of the five tasks in our benchmark and the data processing steps to allow for the complete reconstruction of our results.

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

APPENDIX

## A  THE USE OF LARGE LANGUAGE MODELS (LLMs)

The authors drafted all content in this paper. A large language model was used to assist with grammar, spelling, and language refinement for clarity. The intellectual contributions are entirely the authors' own.

## B  MORE DETAILS OF THE GROUNDING AGENT

We build a multi-agent system composed of a meta-retriever, an LLM-based text interpreter, a structure parser, and a structure memory bank to provide initial result and interact with human annotator to get refined annotation. Specifically, the meta-retriever is built using PubChem APIs Kim et al. (2024) and is responsible for collecting molecular names, properties, structures, and descriptions. The text interpreter leverages large language models (LLMs) to perform named entity recognition and relationship analysis based on the collected meta data. The structure parser is developed using RDKit[1] and handles substructure retrieval, comparison, and validation. The structure memory bank

---

[1]RDKit: Open-source cheminformatics, https://www.rdkit.org

records the refined result of the name-structure mapping by the experts. These three agents work collaboratively to perform structurization on each molecule and output initialized CNER, BNSM, SRG, SFA, and RSL results.

The detail process is given a caption and a molecule, the grounding agent executes a series of steps for structurization, as outlined in Algorithm 1. The text interpreter first performs a CNER task on the caption to extract substructure names. For each extracted name, the meta retriever gathers relevant metadata and supplies it to the text interpreter as examples or contextual information for in-context learning, enabling the conversion of substructure names into molecular representations. The structure parser then validates and locates each substructure within the input molecule. After all instances of the entities are identified, the text interpreter and the structure parser collaborate to ground relational information, update spatial locations, and compute frequencies.

## C  Human Annotation Details and Guidelines

Annotation was conducted by seven undergraduate annotators, all of whom possess a basic knowledge of chemistry and are familiar with molecular naming conventions and structural representations. Prior to the formal annotation process, a briefing session was held to introduce the grounding guidelines and demonstrate the use of the annotation tool. An example of the RSL annotation interface is shown in Figure 5. Additionally, a dry run was conducted to ensure each annotator could correctly operate the tool during the annotation process. To further ensure the quality and reliability of the annotations, all results were subsequently verified by two chemistry experts engaged in molecular research, each with at least six months of professional experience. The error rate between the initial annotation and expert verification was 3.84%, and the final results were generated based on mutual agreement between the two experts.

### C.1  Annotation Guidelines

This section outlines the main annotation guidelines used to verify the outputs of each task. The annotation task is divided into six components, each focusing on a specific aspect of the grounding output.

1. **Molecular Caption Verification**

   **Input:** Caption and Molecule.

   **Objective:** Ensure that the caption accurately describes the corresponding molecule. If any inaccuracies or ambiguities are present, the annotator should revise the caption to provide a faithful and precise molecular description.

2. **Chemical Named Entity Recognition (CNER)**

   **Input:** Caption and CNER result.

   **Objective:** Confirm that all relevant chemical entities mentioned in the caption are correctly identified and extracted. Their corresponding roles are given.

3. **Name-Structure Mapping (NSM)**

   **Input:** Name-to-structure and structure-to-name results.

   **Objective:** Verify the correctness and the uniqueness of the mappings. External reference sources such as PubChem or the MolGenie Ontology are available in the tool to assist in this verification.

4. **Referential Substructure Localization (RSL)**

   **Input:** The predicted locations of a specific entity and the caption.

   **Objective:** Ensure that the locations correctly correspond to the entity mentioned and are contextually grounded.

5. **Substructure Frequency Analysis (SFA)**

   **Input:** A frequency of a specific entity in a molecule, the entity, a caption, and the molecule.

   **Objective:** Validate the total number of localized instances for the entity, ensuring it matches its real frequency in the molecule or the frequency implied or stated in the caption.

**MolGround Annnotation Tool---RSL**

```
=================================================
Task 901: PBDTDTffBT—R
=================================================

Caption: The molecule features eight aromatic and heteroaromatic rings, including
two thiophene rings,  one benzene ring, one benzothiadiazole unit, one benzo[1,2—
b:4,5—b']dithiophen unit, creating an extensively conjugated and aromatic system.

Extracted Entity Name: one benzene ring
NSM: C1=CC=CC=C1
SFA: 1
SRL: [68, 69, 70, 71, 81, 82]

Do you want to edit any of the following attributes?
0. No changes
1. Entity Name
2. NSM
3. SFA
4. SRL
5. Change the caption sentence
Enter the numbers of the attributes you want to edit, separated by commas (e.g.,
1,3).
```

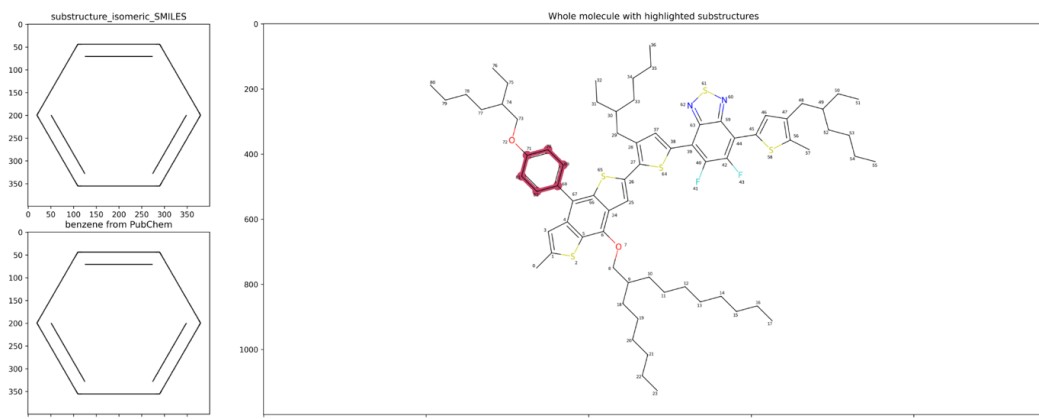

Figure 5: The interface of the Molground Annotation Tool on RSL task.

6. **Substructure Relations Grounding (SRG)**

   **Input:** a predicted relation and the localizations of two instances.

   **Objective:** Verify that the relationship is both contextually and structurally accurate. Let $A$ and $B$ represent a pair substructure instance. The following are the definitions of the five supported relationship types:

   - **BondAttachment:** $A$ and $B$ are directly attached via a single bond between an atom in $A$ and an atom in $B$, without sharing any atoms.
   - **AtomAttachment:** $A$ and $B$ share exactly one atom and are not connected by an explicit bond.
   - **Containment:** $A$ is contained in $B$ if every component of $A$ is also in $B$, and $A$ is not equal to $B$.
   - **Fusion:** $A$ and $B$ are fused if they are both rings and share two adjacent atoms and the bond between them (e.g., naphthalene).
   - **Others:** Any relationship not covered by the above definitions.

   Annotators should determine which of these relationships are present and ensure they are accurately reflected in the grounding output.

---

**Algorithm 1** Grounding Agent

---

**Input:** caption $\mathcal{X}$, molecule $\mathcal{G}$, current memory bank $\mathcal{B}$
**procedure** GROUNDINGAGENT($\mathcal{X}, \mathcal{G}, \mathcal{B}$)
  **Initialize:** $\mathcal{N}, \mathcal{S}, \mathcal{L}, \hat{\mathcal{L}}, \mathcal{K}, \mathcal{F}, \mathcal{R} \leftarrow \emptyset$
  $\mathcal{N} \leftarrow$ TextInterpreter.CNER($\mathcal{X}$)
  $\mathcal{R} \leftarrow$ TextInterpreter.Relation($\mathcal{X}$)
  **for** each $n_j \in \mathcal{N}$ **do**
    $meta \leftarrow$ MetaRetriever.Collect($n_j, \mathcal{B}$)
    $s_j \leftarrow$ TextInterpreter.Name2Struct($n_j, meta$)
    **if** StructureParser.Valid($s_j$) **then**
      $\mathcal{S} \leftarrow \mathcal{S} \cup \{(n_i, s_j)\}$
      $\mathcal{B} \leftarrow \mathcal{B} \cup \{(n_i, s_j)\}$
      $g_i \leftarrow$ StructureParser.Parse($s_i$)
      $\hat{\mathcal{L}}_i \leftarrow$ StructureParser.Retrieve($\mathcal{G}, g_i$)
      $\hat{\mathcal{L}} \leftarrow \hat{\mathcal{L}} \cup \{(n_i, \hat{\mathcal{L}}_i)\}$
    **end if**
  **end for**
  **for** each $(n_i, n_j, r_i) \in \mathcal{R}$ **do**
    $\hat{l}_i \leftarrow \hat{\mathcal{L}}$
    $\hat{l}_j \leftarrow \hat{\mathcal{L}}$
    **for** each $ins_p \in \hat{l}_i$ **do**
      **for** each $ins_q \in \hat{l}_j$ **do**
        $k_{p,q} \leftarrow$ TextInterpreter.Relation($ins_p, ins_q$)
        $\mathcal{K} \leftarrow \mathcal{K} \cup \{(ins_p, ins_q, k_{p,q})\}$
      **end for**
    **end for**
    $\mathcal{L}_i, \mathcal{L}_j \leftarrow$ TextInterpreter.RelationCheck($n_i, n_j, K, r_i$)
    $\mathcal{L} \leftarrow \mathcal{L} \cup \{(n_i, \mathcal{L}_i)\}$
    $\mathcal{L} \leftarrow \mathcal{L} \cup \{(n_j, \mathcal{L}_j)\}$
    $f_i \leftarrow$ Length($\mathcal{L}_i$)
    $f_j \leftarrow$ Length($\mathcal{L}_j$)
    $\mathcal{F} \leftarrow \mathcal{F} \cup \{(n_i, f_i)\}$
    $\mathcal{F} \leftarrow \mathcal{F} \cup \{(n_j, f_j)\}$
  **end for**
  **Output:** $\mathcal{N}, \mathcal{S}, \mathcal{L}, \mathcal{F}, \mathcal{K}, \mathcal{B}$
**end procedure**

---

## D  MOLGROUND DATA SPLIT AND DISTRIBUTION ANALYSIS

We split the QA data into training, validation, and test sets using an 80%/10%/10% ratio for each task, ensuring that no individual molecule appears in more than one split. To evaluate potential distribution shifts among these subsets, we performed similarity-based analyses using Morgan fingerprints. Specifically, we computed the mean pairwise Tanimoto similarity of the molecules within and across the subsets. The average similarities are as follows: train–train (0.304), val–val (0.309), test–test (0.311), train–val (0.302), train–test (0.304), and val–test (0.303). These consistent values indicate that, in terms of fingerprint-based chemical similarity, the datasets are well-matched and exhibit no significant distributional divergence. Furthermore, we visualize the molecular distributions of a random subset using t-SNE embeddings of the fingerprint space, as shown in Figure 6. The

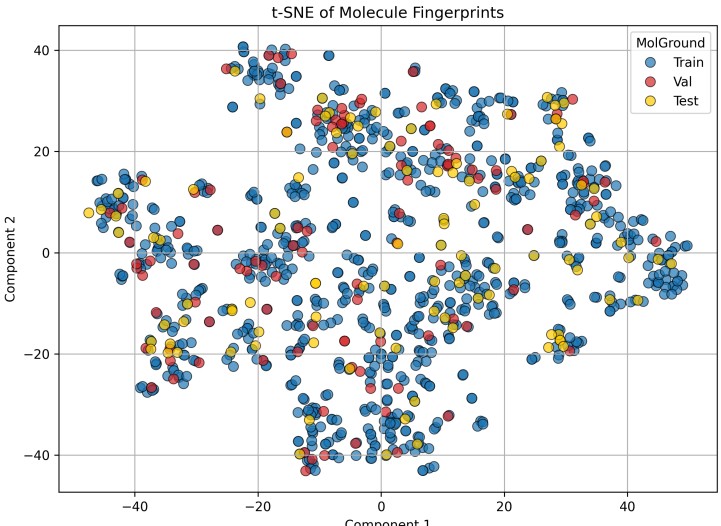

Figure 6: MolGround Data Distribution Visualization.

significant overlap among the train, validation, and test sets in the 2D projection further supports the conclusion that the splits are in-distribution.

## E  MOLECULAR CAPTIONING PROMPT TEMPLATES

The following lists the templates we use to generate molecular captions. After obtaining the caption using GTP4o, the captions are annotated by the annotators. The average similarity of sentence embeddings between the expert caption and ($f_{pub}, f_{general}, f_{specific}, f_{summarize}$) are (0.5254, 0.7115, 0.7630, 0.9906). The summarized caption show the highest similarity with the expert one.

$f_{general}(IUPAC, SMILES)$: Given a molecular IUPAC name and its SMILES, your task is to provide a detailed description, including Basic Structure, Functional Groups, Stereochemistry, Molecular Size and Shape, Physicochemical Properties, Reactivity, Safety and Environmental Impact, etc.

$f_{publication}(literature)$: Given a molecular literature, extract the following information from the literature: 1) Physicochemical Properties: includes physicochemical characteristics of the molecule such as hole mobility, molecular weight, solubility, boiling point, melting point, pKa value (acid dissociation constant), and logP (lipophilicity); 2) Safety Information: Provides information regarding the safety of the molecule, such as its toxicity, carcinogenic, teratogenic, or mutagenic properties. 3) Application Areas: Provides an overview of the applications of the molecule. 4) Spectroscopic Properties: include spectroscopic data of the molecule, such as UV-visible absorption spectrum, infrared spectrum, nuclear magnetic resonance spectrum, and mass spectrometry data

$f_{specific}(StructureImage, SMILES)$: Given a molecular structure image and SMILES, generate a detailed molecular description (within 100 words) focusing number of rings, their types, and associated properties.

$f_{summarize}(f_{general}, f_{paper}, f_{specific})$: Given a molecular structure image, SMILES, IUPAC and three initial descriptions, summarize them and generate a molecular description focusing on basic structure, how substructures connect, their properties, and Stereochemistry. following the example provided below. *Examples*

## F QA TEMPLATES

We list the QA templates used to transform the annotations into QA pairs for each task. For each question, we provide both multiple-choice and open-ended formats. Some representative QA examples are included in `https://anonymous.4open.science/r/MolGround-2025/`.

**CNER task**

CNER Template:
**Question**: "What chemical named entity names are mentioned in the following caption and what are their roles (donor or acceptor)?
Caption: {Caption}
**Answer**: {all chemical named entity names in the caption}.

**BNSM task**

Name-to-SMILES Template:
**Question**: "What is the SMILES representation of the molecular entity {name}?",
**Answer**: {SMILES}

SMILES-to-Name Template:
**Question**: "What is the common or IUPAC name of a compound with the SMILES string {SMILES}?",
**Answer**: {names}

Name-to-InChI Template:
**Question**: "What is the InChI representation of the molecular entity {name}?",
**Answer**: {InChI}

InChI-to-Name Template:
**Question**: "What is the common or IUPAC name of a compound represented by the InChI name {InChI}?",
**Answer**: {names}

Name-to-IUPAC Template:
**Question**: "What is the IUPAC name of the molecular entity {name}?",
**Answer**: {IUPAC}

IUPAC-to-Name Template:
**Question**:"What is the common name of a compound represented by the IUPAC name {IUPAC}?",
**Answer**: {names}

Name-to-Formula Template:
**Question**: "What is the molecular formula for the molecular entity {name}?",
**Answer**: {Formula}

**SFA QA**

SFA-substructure Template:
**Question**:How many instances of the substructure {name} are present in the given molecule, considering the provided context?
Given molecule: {molecule}
Given context: {caption}
**Answer**: {number}

SFA-heteroatom type Template:
**Question**:"How many types of heteroatoms are present in the given molecule?
Given molecule: {molecule}
**Answer**: {number}

SFA-ring Template:
**Question**: "How many {name} rings are present in the given molecule?
Given molecule: {molecule}
**Answer**: {number}

SFA-monocyclic ring type Template:
**Question**: "How many types of monocyclic rings are present in the given molecule?
Given molecule: {molecule}
**Answer**: {number}

SFA-atom Template:
**Question**: "How many {name} atoms are present in the given molecule?
Given molecule: {molecule}
**Answer**: {number}

SFA-non exist ring Template:
**Question**: "How many {name} rings are present in the given molecule?
Given molecule: {molecule}
**Answer**: 0

**SRL QA**

SRL Template:
**Question**: "Based on the caption and molecule provided below, what are the instance-level pairwise relationships for the chemical named entities mentioned in the caption?"
Given molecule: {molecule}
Given caption: {caption}"
**Answer**: {
{"instance1"}: XXXX,
{"instance2"}: YYYY,
{"relation"}: {relation}
}

**S-RSL QA**

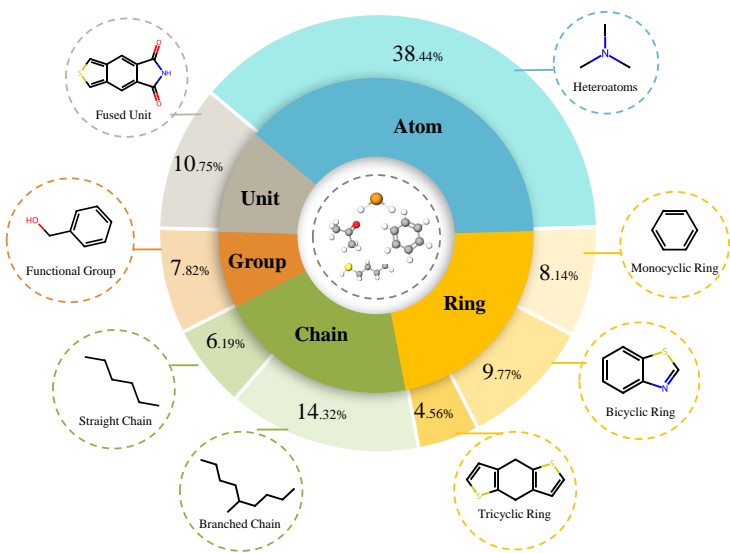

Figure 7: An overview of the MolGround substructure distribution. Five types of structures covers the inner circle, and in the outer circle we list the finer-grained substructure.

Template:
**Question**: "Restricted to the information in the following caption, where are the {name} located in the molecule?" provide their locations as 0-based atom indices.
Given molecule: {molecule}
Given caption: {caption}
**Answer**: {locations}

**M-RSL QA**

Template:
**Question**: "According to the following caption, where are the chemical named entities located in the given molecule?" provide their locations as 0-based atom indices.
Given molecule: {molecule}
Given caption: {caption}
**Answer**: {
{"name1"}: [locations1],
{"name2"}: [locations2],
...
}

Table 4: Performance on Molecular Captioning.

| Model | BLEU-1 | BLEU-2 | BLEU-3 | BLEU-4 |
|---|---|---|---|---|
| GPT-4 | 30.861 | 15.683 | 8.032 | 4.199 |
| GPT-4 + Grounding | **31.178** | **16.698** | **9.002** | **5.004** |

Table 5: Performance on Molecular Classification.

| Model | Aim. ↑ | Cov. ↑ | Acc. ↑ | Abs.T ↑ | Abs.F ↓ |
|---|---|---|---|---|---|
| ATC-CNN | 67.86 | 66.65 | 65.04 | 60.65 | **3.83** |
| ATC-CNN + Grounding | **70.15** | **71.71** | **67.81** | **61.49** | 4.18 |

Table 6: RSL performance with more metrics. Both local ($_l$) and global ($_g$) RSL are reported.

| Tasks | | S-RSL | | | | M-RSL | | | | |
|---|---|---|---|---|---|---|---|---|---|---|
| | Metic | $F1_l$ | $IoU_l$ | $IoU_g$ | $Acc_g$ | $F1_l$ | $IoU_l$ | $IoU_g$ | $Acc_g$ | $Cov_s$ |
| **Pre-trained** | GPT-4o | 0.015 | 0.148 | 0.135 | 0.755 | 0.011 | 0.073 | 0.054 | 0.429 | 0.543 |
| | LLaMA 3.1-8B | 0.006 | 0.175 | 0.141 | 0.672 | 0.000 | 0.018 | 0.018 | 0.083 | 0.126 |
| | Mol-Instruct-8B | 0.000 | 0.000 | 0.000 | 0.000 | 0.000 | 0.000 | 0.000 | 0.000 | 0.000 |
| **ICL(Few-shot)** | GPT-4o | 0.017 | 0.059 | 0.143 | 0.291 | 0.270 | 0.454 | 0.399 | **0.779** | **0.905** |
| | LLaMA 3.1-8B | 0.003 | 0.014 | 0.052 | 0.102 | 0.120 | 0.207 | 0.181 | 0.379 | 0.432 |
| | Mol-Instruct-8B | 0.000 | 0.000 | 0.000 | 0.000 | 0.036 | 0.079 | 0.049 | 0.117 | 0.143 |
| **ICL(RAG)** | GPT-4o | 0.174 | 0.361 | 0.337 | 0.415 | 0.171 | 0.301 | 0.290 | 0.369 | 0.718 |
| | LLaMA 3.1-8B | 0.149 | 0.332 | 0.307 | 0.387 | 0.113 | 0.262 | 0.255 | 0.344 | 0.760 |
| | Mol-Instruct-8B | 0.091 | 0.284 | 0.243 | 0.326 | 0.072 | 0.203 | 0.183 | 0.256 | 0.637 |
| **SFT** | LLaMA 3.1-8B | 0.275 | 0.483 | 0.466 | 0.548 | 0.315 | 0.515 | 0.487 | 0.587 | 0.850 |
| | Mol-Instruct-8B | 0.295 | 0.499 | 0.486 | 0.565 | 0.337 | 0.518 | 0.493 | 0.586 | 0.833 |
| **MLLM** | GPT-4o-Vision | 0.004 | 0.052 | 0.054 | 0.332 | 0.000 | 0.001 | 0.001 | 0.012 | 0.016 |
| | LLaVA-Next-7B | 0.020 | 0.174 | 0.112 | 0.737 | 0.000 | 0.001 | 0.001 | 0.004 | 0.005 |
| | LLaMA 3.2-11B-Vision | 0.010 | 0.113 | 0.085 | 0.555 | 0.003 | 0.062 | 0.046 | 0.280 | 0.402 |
| **Grounding Agent** | GPT-4o | **0.630** | **0.685** | **0.647** | **0.933** | **0.541** | **0.566** | **0.546** | 0.776 | 0.818 |
| | LLaMA 3.1-8B | 0.334 | 0.383 | 0.364 | 0.863 | 0.426 | 0.527 | 0.448 | 0.580 | 0.688 |
| | Mol-Instruct-8B | 0.000 | 0.000 | 0.000 | 0.000 | 0.311 | 0.446 | 0.310 | 0.356 | 0.382 |

# G  EVALUATION METRICS

For CNER and SRG we report the F1-score of the multi-entity/multi-instance prediction and ground truth. Named entities are treated as case-insensitive. For BNSM and SFA, we report accuracy. Additionally, in the BNSM task, all valid structure variants of a molecule are considered equivalent during evaluation. For example, for the SMILES, we canonicalize both the predicted and ground-truth SMILES using RDKit before comparison, ensuring that different yet chemically identical SMILES strings are correctly recognized as equivalent. For RSL, We perform both fine-grained- and coarse-levels evaluation and report F1-score, IoU, accuracy and the substructure coverage. For the fine-grained-level evaluation, we evaluate the ground performance of each instance of each sub-structure one by one. As a substructure could have multiple instances in a molecule, we perform Hungarian matching to find the optical matches and evaluate on the best possible matches. Specifically, we compute the node IoU between grounding prediction and its ground truth of a substructure (i.e., local IoU $IoU_l$) and use it as the Hungarian cost function. For the coarse-level evaluation, we treat all the predictions of a substructure as a whole and compare it with the ground truth annotation. Specifically, the molecule is seen as a graph where atom as node and their bonds as edge, and the grounding task is a node binary classification task. Specifically, the nodes belonging to the mentioning substructures should be highlighted (i.e., label=1). Otherwise, they should have the label of 0. Assuming that the ground truth label for a substructure in the molecule with $m$ atoms is $y = [y_1, ..., y_m]$ and the predicted node classification as $\hat{y} = [\hat{y}_1, ..., \hat{y}_m]$, we compute the average accuracy of the node classification as the global evaluation metric as:

$$Acc_g = \frac{\#correctPrediction}{\#atoms} = \frac{|\hat{y}_i = y_i|}{m} \tag{6}$$

We also compute the IoU of the substructure $S = \{a_i | y_i = 1\}$ and the predicted highlight nodes $P = \{a_i | \hat{y}_i = 1\}$ as another global metric:

$$IoU_g = \frac{S \cap P}{S \cup P} \tag{7}$$

Besides, for the multiple substructure grounding task, we also report the substructure coverage rate $Cov_s$.

# H  DETAILED PERFORMANCE OF DOWNSTREAM TASKS

We provide the detailed performances of the molecular captioning and ATC classification in Table 4 and Table 5 .

