# OpenReview forum: "MolGround: A Benchmark for Molecular Grounding"
_ICLR.cc/2026/Conference — ICLR 2026 Conference Withdrawn Submission_

### Official Review · Reviewer_pZ6m · 2025-10-31

**Soundness:** 3
**Presentation:** 3
**Contribution:** 2
**Rating:** 2
**Confidence:** 3

**Summary:**

This paper introduces MolGround, which is a new benchmark for molecular grounding comprising 187k QA pairs across five tasks: Chemical Name Entity Recognition (CNER), Name-Structure Mapping (BNSM), Referential Substructure Localization (RSL), Substructure Relationship Grounding (SRG), and Substructure Frequency Analysis (SFA).

The authors argue that existing molecular understanding approaches focus on "descriptive" perception while neglecting "referential" perception.
Evaluations show that both general-purpose and domain-specific (M)LLMs struggle on these tasks, though fine-tuning and multi-agent strategies offer modest improvements.

**Strengths:**

Most Comprehensive and Large Molecular Grounding Benchmark. MolGround provides 187k QA pairs across five fine-grained tasks (CNER, BNSM, RSL, SRG, SFA), representing a 3× scale increase over existing benchmarks.

**Weaknesses:**

The authors claim existing benchmarks are 90%+ "descriptive" (Table 1), yet this classification is arbitrary and misleading. For example, they classify property prediction as "descriptive" even when it requires identifying specific functional groups responsible for toxicity—a clearly referential task.

Tasks like CNER (named entity recognition), BNSM (name-to-structure conversion), and SFA (substructure counting) are standard problems in cheminformatics that have been studied for decades using rule-based tools (RDKit, OpenBabel), graph algorithms, and specialized ML models. The novelty lies solely in compiling these into a unified QA benchmark and evaluating LLMs—an incremental contribution insufficient for a top-tier venue.

The paper provides no compelling rationale for why LLMs should be used for these tasks when superior alternatives exist. The paper's evaluations show all models achieve poor accuracy on RSL, yet the authors conclude this "remains challenging and warrants further research" rather than questioning whether LLMs are the wrong tool.

The benchmark construction process is opaque and poorly justified. The authors use GPT-4o to generate captions for molecules lacking descriptions, then use the same model (or its derivatives) to evaluate grounding tasks

**Questions:**

How was the benchmark validated? Provide detailed annotation statistics (rejection rates, refinement cycles, error types). Release confusion matrices and disagreement analysis. Make the data and code publicly available.

What happens when models fail? Conduct systematic error analysis. What percentage of RSL predictions are chemically valid? What types of errors dominate?

---

### Official Review · Reviewer_uW7G · 2025-11-01

**Soundness:** 2
**Presentation:** 1
**Contribution:** 1
**Rating:** 2
**Confidence:** 4

**Summary:**

This paper introduces MolGround, a benchmark designed to evaluate referential understanding in molecular language models . The benchmark includes 187k QA pairs across five proposed grounding tasks - Chemical Name Entity Recognition, Bidirectional Name -Structure Mapping, Referential Substructure Localization, Substructure Relationship Grounding, and Substructure Frequency Analysis. The authors claim MolGround is the largest dataset of its kind and demonstrate its use in evaluating various LLMs and MLLMs.

**Strengths:**

- Paper formulation: The five tasks correspond well to different levels of cognitive and structural reasoning.

**Weaknesses:**

- Dataset not accessible: The full dataset is unavailable via the anonymous link; only partial QA examples are shared. This is a major reproducibility issue for a dataset-centric paper.
- Overstated novelty: Similar or larger datasets already exist, notably MolTextQA (> 500 k pairs)[1], 3DMolLM[2], and Mol-Instructions[3] — all of which address molecular grounding or structure-text alignment. MolGround’s size (187 k) and task scope are not clearly superior. These related works must be acknowledged and the differences w.r.t. the proposed dataset should be clarified.
- Writing and structure: The manuscript is hard to follow. Many paragraphs are lengthy and lack focus; several equations (e.g., Eq. 4) are undefined or poorly explained.
- Evaluation limitations:
    - The benchmark is evaluated only on its own QA tasks; no downstream evaluations.
    - Comparisons with Mol-Instructions and other baselines are insufficiently detailed.
- Data generation issues: The dataset uses GPT-4o to generate captions when unavailable, yet there is no validation or quality analysis for correctness or hallucinations.
- Low question diversity: Nearly all QA pairs concern structural aspects (e.g., rings, chains, substructures), with no property-, function-, or activity-based questions, limiting general utility. The number of types of distinct questions is limited, which will complicate learning
- Unclear utility: The primary use cases for MolGround beyond benchmarking on the dataset itself remain vague.

[1] Laghuvarapu, S., Lee, N., Gao, C., & Sun, J. (2025). MolTextQA: A Question-Answering Dataset and Benchmark for Evaluating Multimodal Architectures and LLMs on Molecular Structure–Text Understanding. Journal of Data-centric Machine Learning Research.
[2] Li, S., Liu, Z., Luo, Y., Wang, X., He, X., Kawaguchi, K., ... & Tian, Q. (2024). Towards 3d molecule-text interpretation in language models. arXiv preprint arXiv:2401.13923.
[3] Fang, Y., Liang, X., Zhang, N., Liu, K., Huang, R., Chen, Z., ... & Chen, H. (2023). Mol-instructions: A large-scale biomolecular instruction dataset for large language models. arXiv preprint arXiv:2306.08018.

**Questions:**

1.  Will the complete dataset (molecules, captions, and QA pairs) be publicly released with a permanent identifier (e.g., DOI or HuggingFace repository)?
2.	Data validation: How were the GPT-4o-generated captions validated for chemical accuracy or factual correctness of the **overall**? Were experts involved in verifying those samples?
3.	Comparative scope: How does MolGround fundamentally differ from MolTextQA, 3DMolLM, and Mol-Instructions beyond task naming and data format?
4.	Use cases: Can you provide examples of downstream tasks (e.g., molecular retrieval, activity prediction) where referential grounding demonstrably helps?
6.	Question diversity: Do you plan to expand the benchmark to include property- or function-related questions instead of focusing solely on substructure localization?
7.	The authors could have leveraged PubChem-300k dataset to expand the number of available captions. Why was this not done ?

---

### Official Review · Reviewer_R6Tk · 2025-11-01

**Soundness:** 3
**Presentation:** 3
**Contribution:** 3
**Rating:** 6
**Confidence:** 3

**Summary:**

The paper introduces MolGround, a comprehensive benchmark designed to evaluate referential molecular understanding, i.e., a model’s ability to link textual chemical mentions to explicit structural components. It formalizes five grounding tasks—Chemical Named Entity Recognition (CNER), Bidirectional Name–Structure Mapping (BNSM), Referential Substructure Localization (RSL), Substructure Relationship Grounding (SRG), and Substructure Frequency Analysis (SFA)—covering multiple cognitive levels.
The benchmark contains 187 k QA pairs spanning 55 k molecules and integrates a human-validated, agent-assisted curation pipeline. Empirical results using general, domain, and multimodal LLMs reveal the difficulty of these tasks (e.g., ≤ 0.016 F1 in zero-shot RSL), and the authors further show that grounding features can improve downstream molecular captioning and ATC classification.

**Strengths:**

1. The paper correctly identifies a key blind spot in current molecule-language research: most benchmarks assess descriptive reasoning (captioning, property prediction) rather than referential understanding (“what/where/which”). The motivation is well-grounded in linguistic and cognitive-semantic theories (Frege, Russell, DRT) and draws a clear analogy to visual grounding in vision-language learning


2. The paper proposed a benchmark with scale - 187 k QA items across 307 substructure types, with train/val/test disjoint by molecule to reduce leakage.

3. The grounding agent baseline with subgraph matching offers a valuable reference point and clearly outperforms standard fine-tuning.

**Weaknesses:**

1. Some task definitions are underspecified or inconsistent (e.g., missing codomain specifications for index sets in RSL/SRG; BNSM acronym varies across text).

2. The taxonomy in Table 1 lists BNSM as fully descriptive despite being defined as bidirectional referential mapping; percentages may not align with row sums

3. The paper lacks a unified “Evaluation Protocols” section describing metrics per task (accuracy vs. F1 vs. exact match) and the treatment of multiple-choice vs. open-ended QA.

4. 	No deterministic rule-based baselines (e.g., OPSIN, Open Babel, RDKit SMARTS) for BNSM, RSL, and SFA to establish upper-bound ceilings.

5. Minor formatting mistakes: Equation formatting and notations occasionally deviate from mathematical convention (e.g., use of “7→” vs. “→”, ambiguous set brackets).

**Questions:**

For the ICL vs. SFT vs. multi-agent: How were in-context examples chosen? Were they automatically retrieved or hand-crafted?

What are the exact heuristics for subgraph matching and constraint enforcement (e.g., tolerance for tautomeric or symmetric structures)?

---

### Official Review · Reviewer_qXFd · 2025-11-03

**Soundness:** 3
**Presentation:** 3
**Contribution:** 3
**Rating:** 4
**Confidence:** 3

**Summary:**

This paper presents MolGround, a large-scale benchmark for evaluating molecular grounding—the ability of language models to align chemical text descriptions with specific molecular substructures. It introduces five tasks covering entity recognition, name–structure mapping, substructure localization, relationship grounding, and frequency analysis. The dataset is built through an interactive human–agent collaboration process to ensure both scalability and quality, resulting in the largest benchmark to date for fine-grained molecular understanding.

**Strengths:**

1. The idea of grounding language in molecular structures is impactful, addressing a crucial gap between symbolic chemical language and structural understanding. The paper targets a clear and underexplored challenge in molecular AI, linking text-level descriptions to molecular substructures.
2. The five subtasks cover multiple reasoning levels, from entity recognition to relational grounding, offering a holistic evaluation framework.
3. The proposed human–agent interactive process smartly balances automation and expert validation.
4. The paper is well-written, with strong figures (e.g., Fig. 1) that clearly explain the workflow.

**Weaknesses:**

1. Although the human–agent collaboration process is clearly demonstrated, it’s difficult to ensure the benchmark’s annotation accuracy and consistency without quantitative validation.
2. The paper lacks diversity analysis to show molecular variety would make the benchmark’s coverage more convincing.
3. The contribution is primarily in dataset design; more discussion on how this framework could inspire model training objectives would strengthen the work.

**Questions:**

please refer to the cons

---

### Note · Authors · 2025-11-25

I have read and agree with the venue's withdrawal policy on behalf of myself and my co-authors.